# A fault diagnosis method for rotating machinery components based on enhanced YOLO v8 and integrated attention mechanism

Xiaoguang Wang[1,2,3], Laohu Yuan [ID][2]*, Le Ma[1,2], Jiafu Liu[4]

1 School of Aviation, Anyang Vocational and Technical College, Anyang, China, 2 School of Aeronautics and Astronautics, Shenyang Aerospace University, Shenyang China, 3 College of Aeronautical Engineering, Anyang University, Anyang, China, 4 School of Aeronautics and Astronautics, Sun Yat-sen University, Shenzhen, China

* ylhhit@126.com

## Abstract

Accurate fault diagnosis of rotating machinery components is the key to ensuring the safe operation of the mechanical system. Aiming at problems such as inaccurate detection of small target fault features and loss of fault information in the process of advanced feature extraction that exist in conventional machine learning and traditional deep learning methods in the fault diagnosis of rotating machinery, this paper improves the YOLO v8. It proposes a YOLO v8-C-OD fault diagnosis method. Firstly, the original vibration signal is transformed into a time-frequency image with enhanced fault features using continuous wavelet transform (CWT) based on Morlet wavelet as the optimal wavelet basis (WBF) to form an experimental sample. Secondly, the C2F module in YOLO v8 Backbone is improved by introducing Omni-dimensional Dynamic Convolution (ODConv) into C2F to dynamically adjust the weight of each convolution kernel to enhance the fault feature extraction capability. Finally, the Convolutional Block Attention Module (CBAM) is fused into YOLO v8 to suppress unnecessary features and significantly improve the model fault diagnosis rate. Finally, experiments conducted on datasets such as those from Case Western Reserve University (CWRU) and Jiangsu Qianpeng Diagnostic Engineering Co. show that the fault classification diagnosis accuracy of this method reaches 100% and 99.75%, respectively, which outperforms existing state-of-the-art models. This study introduces a novel fault diagnosis method and provides a valuable reference for the fault diagnosis of aircraft and industrial rotating machinery components.

## 1. Introduction

Aircraft, automobiles, machine tools, etc, are precise, complex, and highly integrated electromechanical systems, in order to ensure their efficient and safe operation, in which rotating mechanical parts are indispensable. Once the mechanical components

**Data availability statement:** The data for this study can be found in the following repositories: CWRU dataset from Case School of Engineering: https://engineering.case.edu/bearingdatacenter/download-data-file. Paderborn University Bearing Dataset and Gearbox fault simulation test dataset:All relevant data are within the manuscript and its Supporting Information files.

**Funding:** XG W:Grant No:2023C01GX043, the Key Research and Promotion Project of Anyang City (Science and Technology Tackle). The funders had no role in study design, data collection and analysis, decision to publish, or preparation of the manuscript. XG W: Grant No:23A590005, the Key Research Project of Henan Province Universities in 2023.The funders had no role in study design, data collection and analysis, decision to publish, or preparation of the manuscript.

**Competing interests:** The authors have declared that no competing interests exist.

fail, it will lead to serious losses in personnel and the economy. Taking aircraft as an example, according to statistics, the causes of aviation accidents are mainly divided into five categories: pilot operation, mechanical failure, weather causes, and sabotage. Although the proportion of aviation accidents due to mechanical failure has slightly decreased compared with that of the 1950s and 1960s, the proportion is still as high as 23% [1]. In recent years, the fault diagnosis of aircraft electrical systems has been a hot research topic among scholars [2]. For example, the fault diagnosis of aircraft engines [3], aircraft power systems [4], and rudder systems [5]. However, limited research has focused on the fault diagnosis of rotating mechanical components in aircraft. Therefore, it is crucial to develop highly integrated performance fault diagnosis methods for these components to accurately assess their health status and ensure the efficient and safe operation of aircraft.

The efficient and secure operation of a mechanical system depends, to a large extent, on the health of its rotating mechanical components. The information about rotating mechanical components of mechanical systems collected under various health states often contains rich data, but these data are characterized by nonlinearity, weakness, and non-smoothness. Therefore, analyzing different health states and accurately diagnosing fault states remains a complex problem. At present, the acquisition of fault signals for rotating machinery components mainly includes temperature, sound, vibration, and oil samples. Among them, the majority of researchers have adopted vibration signals to realize the fault diagnosis of rotating machinery components because of their advantages of being easy to acquire and analyze [6].

When it comes to diagnosing problems with spinning equipment parts, some scholars have proposed methods such as ensemble empirical modal decomposition [7], empirical mode decomposition [8], wavelet transforms [9], and other methods based on the original vibration signals for fault feature extraction of rotating machinery components. Peng Chaoqin et al. [10] extracted the temporal features of each type of sensor signal pair with channel signals through a gated recurrent unit (GRU), based on which a multi-channel attention mechanism was introduced to autonomously fuse the features of different channels. Finally, a classifier was utilized to achieve fault diagnosis. A noise-assisted technique was presented by Wang et al. [11], who based their work on the conventional empirical mode decomposition of EMD. This technique combined the fault-related modes with varying noise levels in a nonlinear and adaptive manner. By doing so, it suppressed the fault-irrelevant components. and ultimately improved the diagnostic accuracy. For the non-stationary and nonlinear difficulties of rotating equipment bearing vibration signals, Ben Ali et al. [12] suggested a feature extraction technique that combines energy entropy with empirical modal decomposition (EMD). Key characteristics were extracted from the vibration signals using this technique. An artificial neural network (ANN) uses these properties for training and learning. An effective diagnosis of bearing faults is finally realized. Zhang et al. [13] transformed the raw dataset into a spectrogram format to preserve the original time-domain signal information to a greater extent. Following this, the converted dataset was trained using a deep fully convolutional neural network to intelligently classify vibration signals with various defect locations and

damage degrees. It is finally possible to diagnose rolling bearing problems. Zhu Yungui et al. [14] used the de-phasing algorithm (de-phasing algorithm, DPA) to suppress strictly periodic components such as rotational frequency and its harmonics. Then, the fault impact component is enhanced by multipoint optimization of the minimum entropy inverse pleated product, and the enhanced signal is subjected to spectral analysis for extracting the fault characteristics and accomplishing the fault diagnosis. Sha Yundong et al. [15] used wavelet packet decomposition (WPD) of vibration signals to obtain several nodal components and screened and reconstructed each nodal component with multi-parameter fusion, such as skewness value (Skew). Finally, fault characterization was performed by envelope demodulation. However, these methods are based on the processing of raw vibration signals, which cannot process large data batches for better readability, and in the diagnosis process, they rely heavily on the experience of experts, resulting in limitations in their application receiving limitations.

In recent years, deep learning has advanced dramatically in fields like computer vision and natural language processing. As a result, it has also started being used in machinery fault diagnosis. Specifically, in this application, the Convolutional Neural Network (CNN), a well-known illustration of deep learning, exhibits excellent performance in image feature extraction and is widely used in machinery fault diagnosis. Ye Zhuang et al. [16] utilized empirical mode decomposition (EMD) to process the vibration signals, and the fault features of the obtained multichannel, one-dimensional vibration signals were transformed into an image input. Then, a dynamic receptive field is used in the convolutional layer to extract the details of the multi-channel image features comprehensively, and the channel fault features are enhanced by the crag-weighted multi-channel fusion method. Finally, the gearbox fault diagnosis is achieved via simulation experiments. Zhou et al. [17] used wavelet packet decomposition and reconstruction techniques to eliminate the noise from the original bearing vibration signal. The processed vibration signal is then converted into a two-dimensional time-frequency picture using a frequency-sliced wavelet transform. Finally, the 2D time-frequency image is used to train a deep neural network based on ResNet50 to identify errors. Zhu et al. [18] utilized the wavelet packet transform (WPT) to extract time-frequency features from bearing signals, forming a time-frequency feature matrix. Sensitive fault features were then extracted from the time-frequency feature matrix produced by WPT using Multi-Weighted Singular Value Decomposition (MWSVD), while non-sensitive features were removed using entropy weight and singular value contribution rate. To diagnose bearing faults, a support vector machine classifier was fed the resultant feature matrix. Chen et al. [19] first divided the bearing vibration signal into time-domain segments, followed by fault signal extraction from the segmented signal using time-domain segmentation. The bearing fault characteristic frequency was subsequently improved using the spectral averaging method, leading to a successful diagnosis of bearing faults. To produce a more consistent entropy value and get over the limitations of the original multi-scale weighted permutation entropy structure, Wang et al. [20] developed a new composite multi-scale weighted permutation entropy (GCMWPE) nonlinear approach. This made it possible to create a highly witnessed levy set and extract bearing characteristics from various scales. The main characteristics of GCMWPE were then extracted using the S-ISO technique. Bearing defects were effectively diagnosed and identified by feeding the MPA-SVM with the combination of GCMWPE and the S-ISO ensemble. First, Liu et al. [21] adaptively calculated the center frequency of defective pulses, reduced the interference of random pulses, and evaluated the energy spectrum distribution of faulty pulses using the variance statistics index. Then, to speed up convergence, the enhanced mayfly optimization method (MMA) searches for the ideal resonant demodulation band. Lastly, the filtered signal's squared envelope spectrum is processed. It is possible to diagnose bearing problems. Yen-Tiger et al. [22]converted the fault data vibration signals into time-frequency images by continuous wavelet transform (CWT), extracted the fault features by using a dense connected convolutional network (DenseNet) model, and finally pre-trained and completed the fault diagnosis by using support vector machines (SVM). The study shows that the transformation of one-dimensional vibration signals into two-dimensional images for fault feature extraction as well as fault diagnosis and identification, has obtained favorable results. However, this type of approach to fault feature extraction, due to the uncertainty and diversity of fault features, results in the loss of key fault features in the actual fault feature extraction, which ultimately affects the fault diagnosis results.

Inspired by the above research, because the original fault information of the rotating parts is collected as vibration signal data, through the detection of fault information targets to achieve the purpose of fault diagnosis, there must be two conditions: first, there is a need for an image conversion method that contains significant fault information characteristics and is easy to detect; and second, there is a need for an advanced technology that can be used to extract the fault characteristics and reasonable classification, ultimately achieving the detection of the target. technology. In summary, this paper puts forward the YOLO v8-C-OD fault diagnosis model. Firstly, the Morlet wavelet is used as the CWT method of WBF to carry out preliminary feature extraction of the original faulty vibration signal data, which is converted into a time-frequency image with feature enhancement, and then the feature image is inputted into the diagnostic model presented in this thesis. By replacing the convolution of the C2F module of the YOLO v8 network using ODconvolution, and at the same time fusing CBAM in its Backbone, to perform fault diagnosis and identify rotating machinery components, the YOLO v8-C-OD fault diagnostic model is designed to recognize and extract the deep characteristics of the fault information. The model has an important reference value for fault diagnosis research in aircraft, industrial rotating mechanical components, and related fields. The contributions of this paper are as follows:

1. A new YOLO v8-C-OD fault diagnosis model based on the YOLO v8 algorithm for rotating machinery components is proposed and experimented on datasets such as CWRU and Paderborn University Bearing Data to demonstrate the feasibility and effectiveness of the method.

2. A new C2F-OD module is designed, which significantly strengthens the ability of the model to focus on the key features of tiny targets by constructing a dynamic feature fusion mechanism, and effectively solves the problem of insufficient feature characterization in small target detection.

3. Considering that the original YOLO v8 algorithm may introduce blurring or distortion in the dynamic up-sampling module, and the feature information does not match, resulting in the loss of feature extraction. The CBAM attention mechanism is introduced to improve the integrity of fault features through the adaptive weighted fusion of the feature matrix and attention weights, which effectively solves the problem of loss of fault feature extraction in the upsampling process.

## 2. Theoretical foundations

### 2.1. YOLO v8 model

YOLO v8 is a deep learning network that integrates the tasks of target detection, category prediction, and real-time tracking. The network adopts anchor-free detection based on the center detection algorithm, which reduces the number of prediction frames and accelerates the non-maximum suppression (NMS), efficiently raises the target detection accuracy, and is especially friendly to small targets. Meanwhile, the YOLO v8 network, in its neck part, uses features directly connected without enforcing the same channel size, which reduces the parameter counts and the overall size of the tensor to achieve faster detection. In addition, YOLO v8 incorporates a new convolution by changing the first 6x6 convolution of the stem to a 3x3 convolution, and the main building block uses C2f instead of C3, the module shown in Fig 1(a), which enhances the network's potential for learning. Additionally, YOLO v8's detecting head component has been enhanced as well; Fig 1(b) illustrates this component. By decoupling the head structure, the network may concentrate more on various detection tasks, increasing detection accuracy and efficiency.

YOLO v8, an advanced model in the domain of target detection as shown in Fig 2, significantly enhances the accuracy of target detection by introducing a dynamic upsampling module and a feature fusion strategy. However, these methods also expose some limitations. First, the dynamic upsampling module may introduce blurring or distortion while improving the resolution of the feature map, which may affect the accuracy of the features. Second, in the feature fusion stage, direct splicing may lead to mismatches or conflicts of information, which in turn degrade the performance of target detection due to the significant differences in semantics and resolution of features at different scales. At the same time, although this

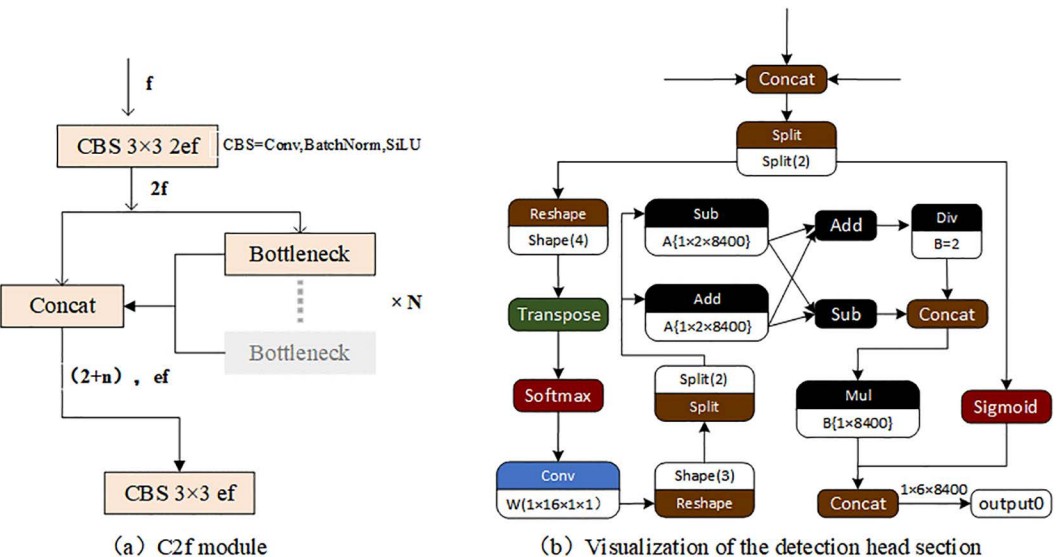

**Fig 1. Structure of YOLO v8 detection head part and C2f module.**

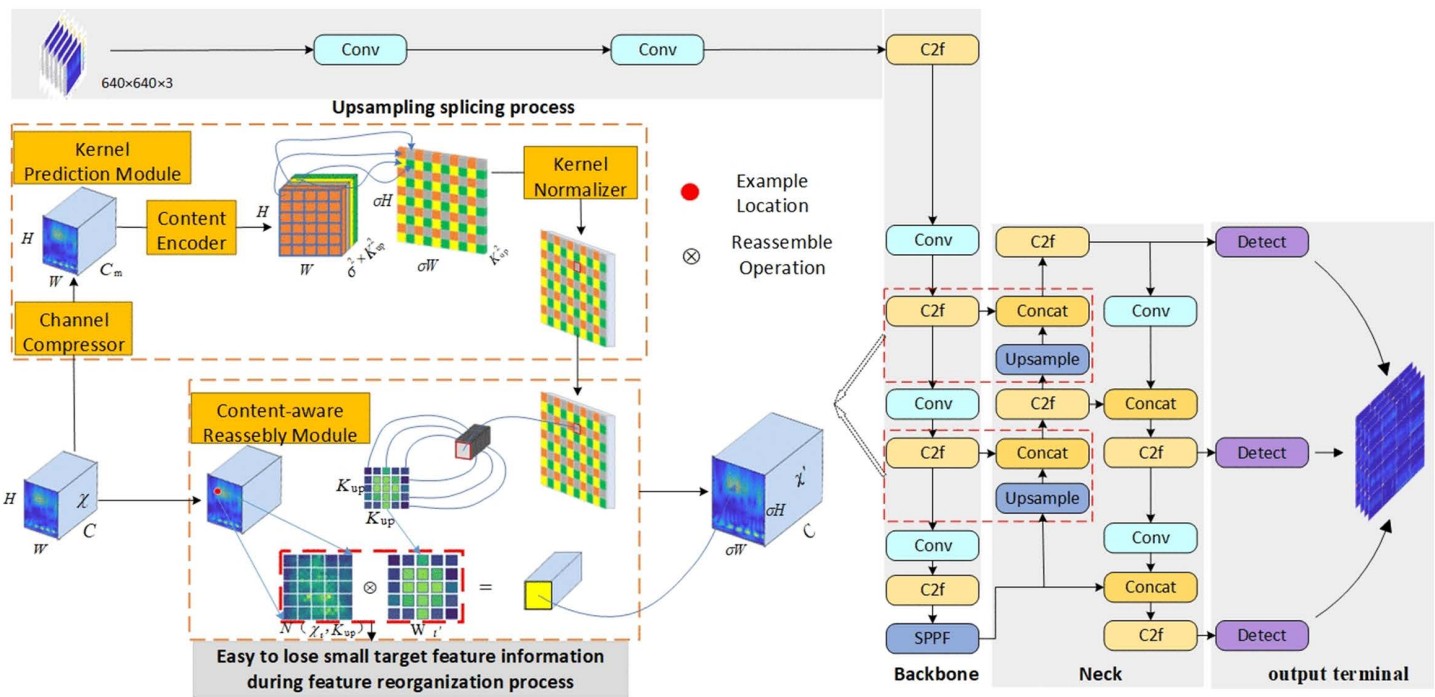

**Fig 2. YOLO v8 network structure.**

network further enhances the learning capability of the network by using C2f instead of C3, it still tends to be overlooked during feature extraction because of the inconspicuous features of slight vibration faults. These challenges are particularly prominent for small-target detection. Since small targets account for a small proportion of the image and have relatively

little feature information, up-sampling and splicing operations may not effectively recover their detailed information, resulting in lower detection accuracy for small targets. In addition, the feature scale of small targets varies over a wide range, increasing the difficulty of overall detection and classification.

To address this problem, this paper introduces the channel and spatial attention mechanisms into the YOLO v8 network during the upsampling and concatenation operations, aiming to reduce blurring and distortion and improve the accuracy of feature extraction, thereby enhancing detection accuracy, especially for small targets.

## 2.2. Channel and spatial attention mechanism

The Attention Mechanism in neural networks is a method for managing computational resources by prioritizing important tasks and mitigating information overload within a limited capacity. By integrating the Attention Mechanism, the model may minimize attention to less important features or ignore irrelevant data while focusing on important information within a huge pool of inputs. This approach effectively tackles the challenge of information overload and enhances the efficiency and accuracy of task processing.

The conventional channel feature maps are abundant in attentional information among pixel points but neglect spatial attention information [23]. The convolutional attention module, on the other hand, is lightweight. The CBAM structure consists of a spatial attention module and a serial channel attention module. Fig 3 illustrates the general structure. By adaptively choosing channel attention and spatial attention, weighting the sum of channel attention and spatial attention, and ultimately generating hybrid feature vectors, the CBAM attention mechanism considers the interaction of the input data in both channel and spatial dimensions. It then integrates cross-channel and spatial information using convolutional operations. To facilitate information flow in the network, enhance network performance and accuracy, and collect more thorough and trustworthy attention information, spatial attention gathers rich spatial information, prioritizes significant features, and suppresses extraneous data [24].

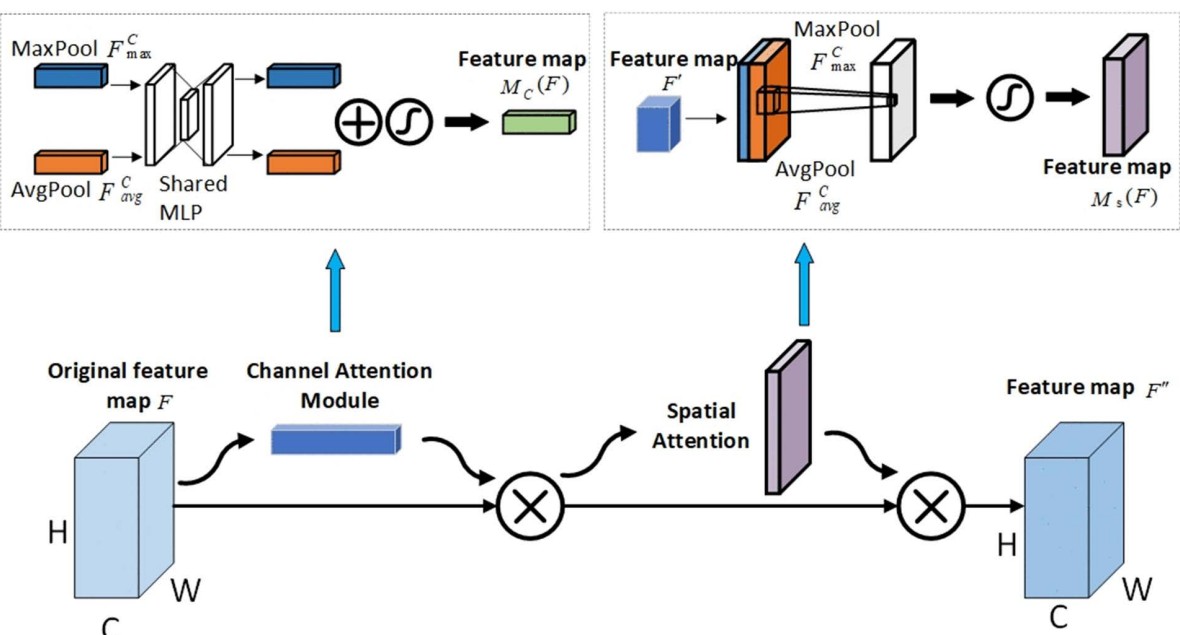

**Fig 3. CBAM Convolutional Attention Mechanism Architecture.**

The CBAM process has the following 2 main operations [25]:

$$F' = M_C(F) \otimes F \tag{1}$$

$$F'' = M_S(F') \otimes F' \tag{2}$$

Eq: $F \in R^{C \times H \times B}$ are the input features, two distinct $1 \times 1 \times C$ average-pooled feature maps and maximum-pooled feature maps are produced by the channel attention module using global maximum pooling and global average pooling. Equation (3) describes the computing process. By fusing the channel's maximally and averagely pooled feature maps, the spatial attention module takes the channel attention module and creates the $H \times W \times 2C$ feature map, and the computational process is shown in Equation (4).

$$M_C(F) = \sigma(MLP(AvgPool(F))) + \sigma(MLP(MaxPool(F)))$$
$$= \sigma(W_1(W_0(F_{avg}^C))) + \sigma(W_1(W_0(F_{max}^C))) \tag{3}$$

$$M_s(F) = \sigma(f^{(7 \times 7)}([AvgPool(F); MaxPool(F)]))$$
$$= \sigma(f^{(7 \times 7)}([F_{avg}^c; F_{max}^c])) \tag{4}$$

Eq:

$M_C \in R^{C \times 1 \times 1}$ —attentional weight on the channel dimension;

$M_S \in R^{1 \times H \times B}$ —attention weight on the spatial dimension;

$\sigma$ — sigmoid function; MLP — shared neural network;

$W_0$ and $W_1$ are the weight of two MLP layers;

$AvgPool$ —Average Pooling;

$MaxPool$ —Maximum pooling;

$f^{7 \times 7}$— $7 \times 7$ convolutional computation;

$F_{avg}^c$— average pooling feature, size $1 \times H \times W$;

$F_{max}^c$ — maximum pooling feature, size $1 \times H \times W$.

### 2.3. The ODConv modules

ODConv (Omni-dimensional Dynamic Convolution) [26–28] is a state-of-the-art dynamic convolution design that enhances feature representation by introducing an attention mechanism in different dimensions of the convolution kernel. The structural flowchart is shown in Fig 4. Unlike the traditional methods that focus on only one dimension of the number of convolution kernels, by introducing an attention mechanism in four dimensions—space, input channel, and output channel—ODConv can dynamically modify each convolution kernel's weight. The feature extraction capability of convolutional neural networks is greatly improved while maintaining a low computational overhead and parameter count, thanks to ODConv's ability to dynamically adjust the weight of each convolutional kernel by introducing the attention mechanism on the four dimensions of space, input channel, output channel, and convolutional kernel, especially in small target detection and fine-grained classification tasks. This approach enhances the model's ability to efficiently acquire input information and capture contextual cues. Its definition is given by Eq. (5) as follows:

$$Y = \left( \sum_{i=1}^{n} \alpha_{\omega i} \, \alpha_{ci} \, \alpha_{fi} \, \alpha_{si} \, W_i \right) * X \tag{5}$$

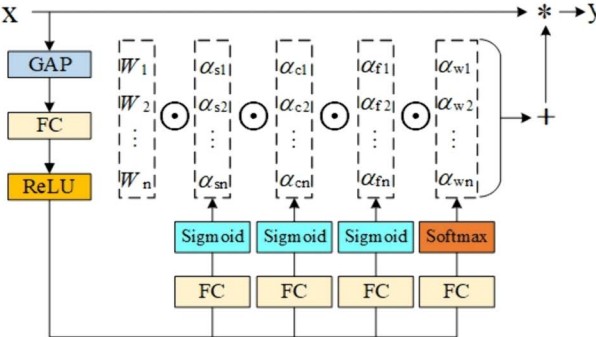

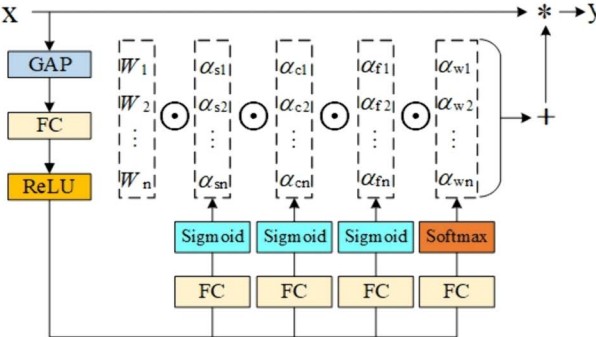

**Fig 4. ODConv structure diagram.**

where X and Y denote the input and output features, respectively, $W_i$ denote the convolution kernel, $\alpha_{\omega i}$, $\alpha_{ci}$, $\alpha_{fi}$, $\alpha_{si}$ denote the convolution kernel dimension, input channel dimension, output channel dimension, and spatial dimension attention weights, respectively, and – denotes the multiplication operation performed on the four dimensions.

In this study, we improve the C2F module in the YOLO v8 Backbone network by replacing the original conv convolution in the C2F module with the ODConv convolution, constituting the C2F-OD module, as shown in Fig 5, which can realize more selective feature fusion and maintain the feature extraction capability while reducing both the parameter redundancy as well as the model parameters, and also improve the feature extraction capability, thus improving the feature representation capability of the model.

## 3. YOLO v8-C-OD model

In this paper, we incorporate the attention mechanism into the YOLO v8 network with an improved C2F module and propose a fault diagnosis method based on the YOLO v8-C-OD model for the diagnosis of problems related to faults in rotating machinery components.

First, the raw vibration signals of the faulty rotating machinery components acquired are divided into signal segments. To obtain as many samples as possible, we divide the original vibration signals by overlapping a part of the data points between the front and back neighboring segments, and the specific procedure is presented in Fig 6.

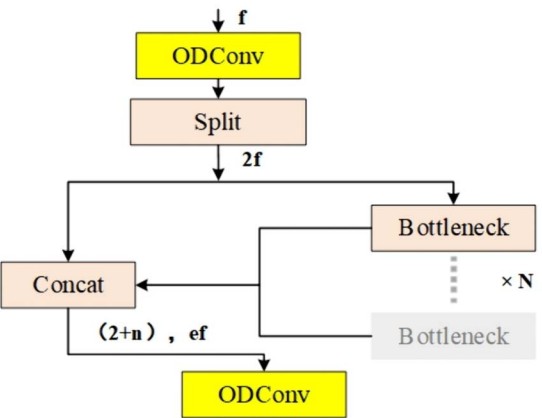

**Fig 5. C2F-OD model.**

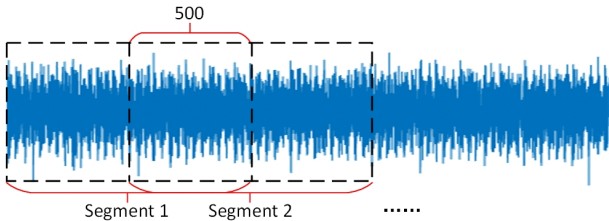

**Fig 6. Data segmentation.**

The original oscillation signals from the moving components are transformed into time-frequency representations with improved characteristics using a Morlet wavelet-based continuous wavelet transform (CWT), which is selected as the optimal wavelet basis function (WBF). Subsequently, features are extracted from the vibration signals using the YOLOv8 network with the improved C2F module combined with the CBAM. Finally, the classification capability of the YOLOv8 network is utilized to conduct fault diagnosis of the rotating components. The entire process of fault diagnosis is illustrated in Fig 7.

### 3.1. Signal-to-image conversion

The wavelet transform is well-suited for detecting transient anomalous signals entrained in normal signals, distinguishing between bursty and stable signals, and determining the state and location of their energy distributions. The continuous wavelet transform (CWT) is capable of decomposing the original signal into a time-scale plane with localized components represented by scaling and translation operations. Therefore, the CWT can be used during preprocessing to obtain a sensitive representation of the signal, which is expressed as follows:

$$W_{\varphi}(a, b) = \frac{1}{\sqrt{a}} \int x(t)\varphi^*(\frac{t-b}{a})dt, a > 0$$

(6)

formula:
$W_{\varphi}(a, b)$ — The result of the wavelet transform (Wavelet coefficient);
$x(t)$ — Signal to be analyzed;
$\varphi$ — Wavelet basis functions;
$*$ — denotes the conjugate;
a — the amount of expansion (frequency parameter);
b — the amount of translation (time parameter).

Given that the Morlet wavelet is a Gaussian envelope containing a single-frequency sinusoidal function [29], its shape is symmetric and smooth with explicit analytical equations, the similarity coefficients are large, and it can extract more effective signal fault features.

In this paper, the Morlet wavelet is adopted as the CWT method of WBF [30] to carry out preliminary feature extraction of fault data, and the original vibration signal of the rotating parts is converted into a 224 × 224 × 3 time-frequency image with feature enhancement, and the image before and after signal conversion is shown in Fig 8.

### 3.2. Improving YOLO v8

Fig 9 demonstrates the improved fault diagnosis framework YOLO v8-C-OD proposed in this study, which achieves an optimized replacement of the original YOLO v8 anchor mechanism by deeply reconfiguring the target detection paradigm. At the core architecture level, this study innovatively replaces the standard convolutional layer (conv) in the traditional C2F

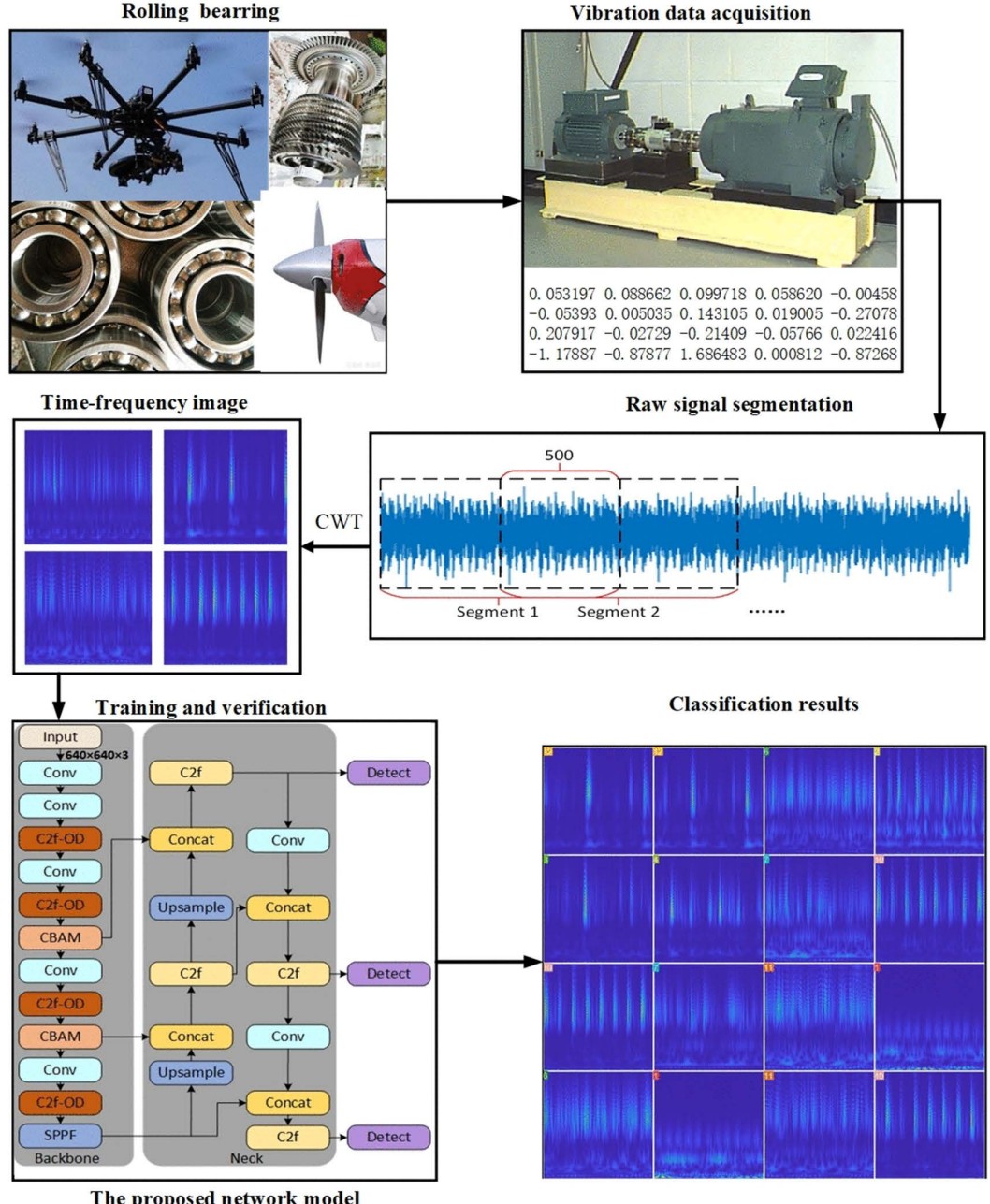

**Fig 7. Fault diagnosis flowchart.**

module with the ODconv structure shown in Fig 5 to form an enhanced C2F-OD module. This improvement significantly strengthens the model's ability to focus on key features of tiny targets by constructing a dynamic feature fusion mechanism, especially in multi-dimensional feature extraction, which demonstrates excellent performance: through cross-channel feature reorganization and spatial dimension feature enhancement, it effectively solves the industry problem of insufficient feature characterization in small target detection. In terms of feature enhancement strategy, the study

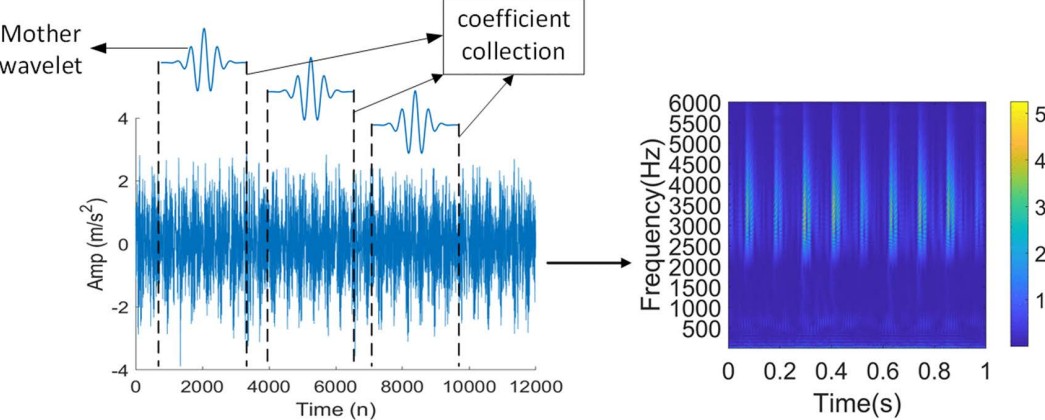

**Fig 8. Image conversion.**

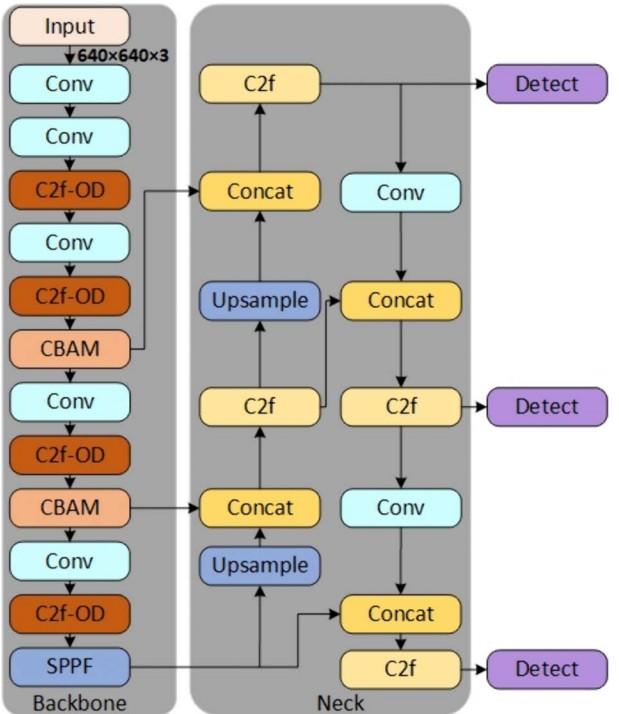

**Fig 9. YOLO v8-C-OD troubleshooting model.**

introduces a lightweight CBAM attention mechanism to realize accurate feature modulation. In the specific implementation, a two-dimensional attention module is embedded in the key nodes of the fourth and sixth layers of the backbone network (Backbone), which independently generates the attention weight map along the channel and spatial axes of the feature tensor through the cascade design of the channel attention sub-module and the spatial attention sub-module. Through the adaptive weighted fusion of feature matrix and attention weights, not only the selective enhancement of important feature channels is realized, but also the fine localization of key spatial regions is accomplished. This

two-dimensional synergistic optimization mechanism significantly improves the representational integrity of the feature pyramid under the premise of effectively suppressing feature redundancy, and effectively solves the problem of loss of high-level feature extraction in the up-sampling process of the YOLO v8 model. The same workstation, a GeForce RTX 3050 graphics card with 4GB of video RAM running Windows 10, is used for both training and testing. Python 3.8 is used to design the model under the Pytorch deep learning framework. Batch = 64, dropout = 0.5, lr0 = 0.001 as the initial learning rate, and dynamic learning rate adjustments were made throughout the training period.

## 4. Experimental comparative analysis

The primary data sources chosen for experimental validation in this paper are the vibration signals of bearing failures at Case Western Reserve University (CWRU) [31,32] and the fault signals of gearboxes from Jiangsu Qianpeng Diagnostic Engineering Company Limited [33]to confirm the effectiveness of the YOLO v8-C-OD fault diagnosis method suggested in this paper for rotating machinery systems. These signals simulate various health condition vibration signals of bearings and gearboxes, which are vibrating components of machinery. The experimental analysis process is described in detail in this section.

### 4.1. Bearing fault diagnosis

**4.1.1. Experimental data set.** The CWRU dataset is a globally recognized standard dataset for diagnosing bearing faults, and much of the existing research relies on this data for experimental validation and a relatively fair comparison with other fault diagnosis methods. Illustrated in Fig 10, A 2 hp motor (to the right), a torque sensor encoder (in the center), a dynamometer (to the left), and control electronics make up the test rig for the experimental arrangement. The SKF 6205−2RS JEM deep groove ball bearing supports the motor shaft. A single point of failure is induced in the test bearing through electrical discharge machining, and the accelerometer is mounted on the base housing. The sampling frequency for the fan-side bearing was 12 kHz, while for the drive-side bearing it was 12 kHz and 48 kHz. Three failure states, inner-ring failure (IRF), rolling element failure (BF), and outer-ring failure (ORF), were simulated using the experimental setup. Bearing vibration signals were collected for a total of 11 fault states under normal conditions and three fault states with different damage diameters.

In this experiment, a sampling frequency of 12 kHz is selected, and the data collected at the drive end under 12 types of health conditions is used to better simulate several types of failures of rotating mechanical components and the difference in the degree of damage to the fault location under different types of failures.

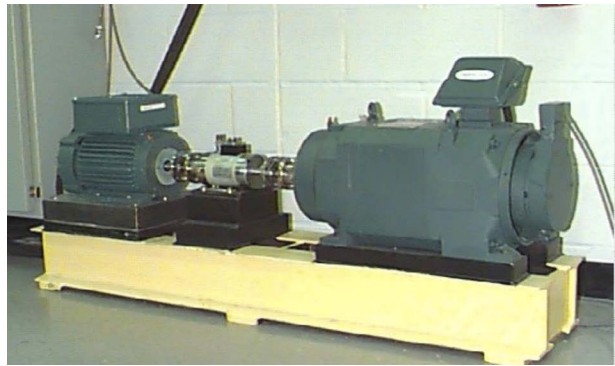

**Fig 10. CWRU test stand.**

**4.1.2. Sample description.** Four loads (0, 1, 2, and 3 horsepower) are included in the dataset for each of the 12 health conditions that were chosen for this experiment, totaling 120,000 data points. To clearly extract periodic fault features in time-frequency analysis, the data window must cover at least 5–10 complete fault impact cycles, providing sufficient feature learning space for subsequent time-frequency transformations or deep learning models. With a sampling frequency of 12 kHz and a total of 120,000 data points, 5 cycles correspond to approximately 674 data points, while 10 cycles correspond to approximately 1,348 points. This paper selects a sample length of 1000 points, covering approximately 7.2 fault cycles. This length offers advantages over longer samples like 2048 or 4096 in terms of GPU memory usage and computational efficiency, while aligning with industrial sliding window design principles (where sample length is typically 1/5–1/3 of the signal variation period, with 1000 points falling within this optimal range). The core data augmentation strategy employs a 50% overlap rate (where adjacent windows overlap by 500 points). This approach significantly enhances the robustness and stability of the features learned by the model by preventing fault impact events from being entirely missed by all windows. Simultaneously, it effectively expands the training dataset size, alleviating the deep learning model's demand for large data volumes. Ultimately, this achieves dual optimization: ensuring the completeness of feature extraction while enhancing the efficiency of model training. Finally, each type of fault data is divided into 239 groups. After applying the continuous wavelet transform (CWT), 239 time-frequency images are obtained for each operational state. The time-frequency images obtained for each fault type under the four load conditions are merged to form 956 time-frequency images for each health condition. Then, in an 8:2 ratio, each image category is divided into training samples and test samples. Finally, as shown in Table 1, the data files from every category are merged to provide a sample set with 11,472 time-frequency pictures. Fig 11(a)-(l) shows the time-frequency pictures for each of the 12 health problems listed in Table 1 in consecutive order.

**4.1.3. Comparative analysis of diagnostic results.** Firstly, the standard YOLO v8 fault diagnosis model and the YOLO v8-C-OD fault diagnosis model constructed in this paper are utilized respectively to train the training samples obtained from the 12 categories of health states constructed above, and then the optimal model obtained in the training stage is utilized to evaluate the test samples, and to determine the final fault diagnosis accuracy, The corresponding confusion matrix and results are illustrated in Fig 12(a)(b).

To minimize the effect of chance and specificity, the experiment is iterated 10 times, and the mean test accuracy is contrasted with established cutting-edge approaches like CNN-HMM [34], TTLN [35], MTSTLF [36], ResNet-SVM [37], DGNN [38], and DenseNet-SVM [22]. In this experiment, we used the CWRU bearing failure dataset to validate each method. In literature [34], by creating a 50×50 matrix as a sample for the raw vibration signal data, a convolutional neural network-based

**Table 1. Description of sample types in the CWRU dataset.**

| Fault types | Fault diameters (mm) | Number of training/test samples | Sample points | Lable |
|---|---|---|---|---|
| Normal status | 0 | 765/192 | 1000 | Class1 |
| Fault in inner race | 0.1778 | 765/192 | 1000 | Class2 |
| Fault in inner race | 0.3556 | 765/192 | 1000 | Class3 |
| Fault in inner race | 0.5334 | 765/192 | 1000 | Class4 |
| Fault in inner race | 0.7112 | 765/192 | 1000 | Class5 |
| Fault in ball | 0.1778 | 765/192 | 1000 | Class6 |
| Fault in ball | 0.3556 | 765/192 | 1000 | Class7 |
| Fault in ball | 0.5334 | 765/192 | 1000 | Class8 |
| Fault in ball | 0.7112 | 765/192 | 1000 | Class9 |
| Fault in outer race | 0.1778 | 765/192 | 1000 | Class10 |
| Fault in outer race | 0.3556 | 765/192 | 1000 | Class11 |
| Fault in outer race | 0.5334 | 765/192 | 1000 | Class12 |

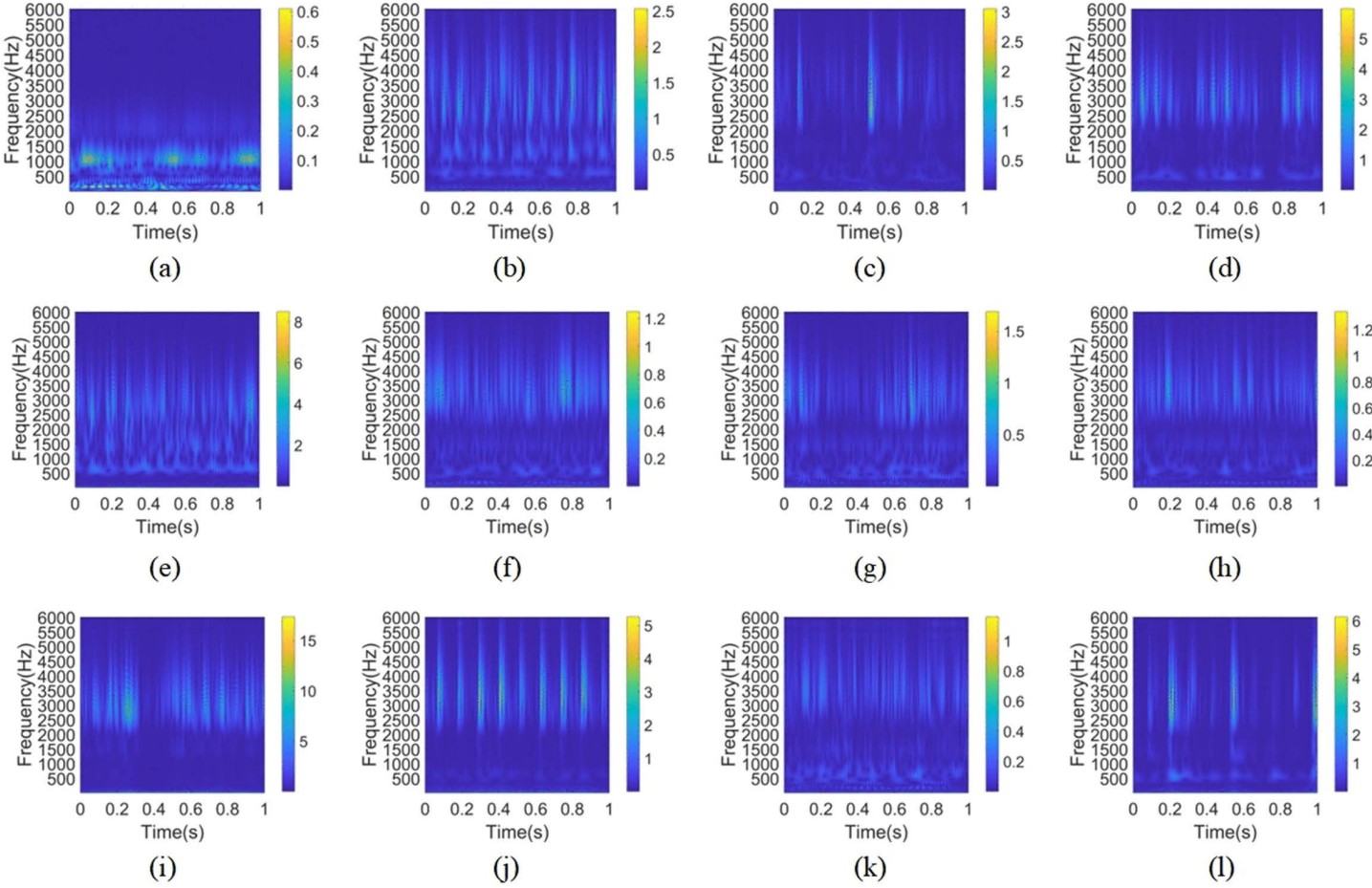

**Fig 11. Time-frequency diagrams of vibration signals for 12 operating conditions of the bearing.** (a) Class 1; (b) Class 2; (c) Class 3; (d) Class 4; (e) Class 5; (f) Class 6; (g) Class 7; (h) Class 8; (i) Class 9; (j) Class 10; (k) Class 11; (l) Class 12.

Hidden Markov Model (CNN-HMM) is suggested for the classification of 12 medical diseases. In literature [35], a Transformer transfer learning network (TTLN) for accurate fault diagnosis under cross-condition scenarios, particularly when target domain data are limited. In the literature [36], the Enhancement of migration learning capability for cross-machine fault diagnosis of rolling bearings by integrating metrics, probability distributions, and geometric similarities of two types of multi-task self-supervised transfer learning frameworks (MTSTLF) and multi-angle feature migration methods. In literature [37], a combination of the pre-trained convolutional neural network RseNet and SVM was used to classify 12 health conditions of rolling bearings. In literature [38], a deep generative neural network (DGNN) was introduced. Using samples of ten medical diseases obtained at 0 horsepower motor loads, the model was trained, and subsequently, a comprehensive classification of 10 health states was performed for operating conditions at 1, 2, and 3 hp motor loads. In literature [22], a combination of densely connected convolutional networks (DenseNet) and SVM was used to classify 12 health conditions of rolling bearings. Table 2 displays the five methods' respective diagnostic outcomes, which are 98.125%, 98.9%, 95.63%, 98.75%, 97.81% and 99.06%.

Compared with the standard YOLO v8 fault diagnosis model and other methods in the literature, the performance of the proposed fault diagnosis method is significantly improved, and the diagnosis accuracy is higher, which proves the feasibility and superiority of the proposed method in the field of bearing fault diagnosis.

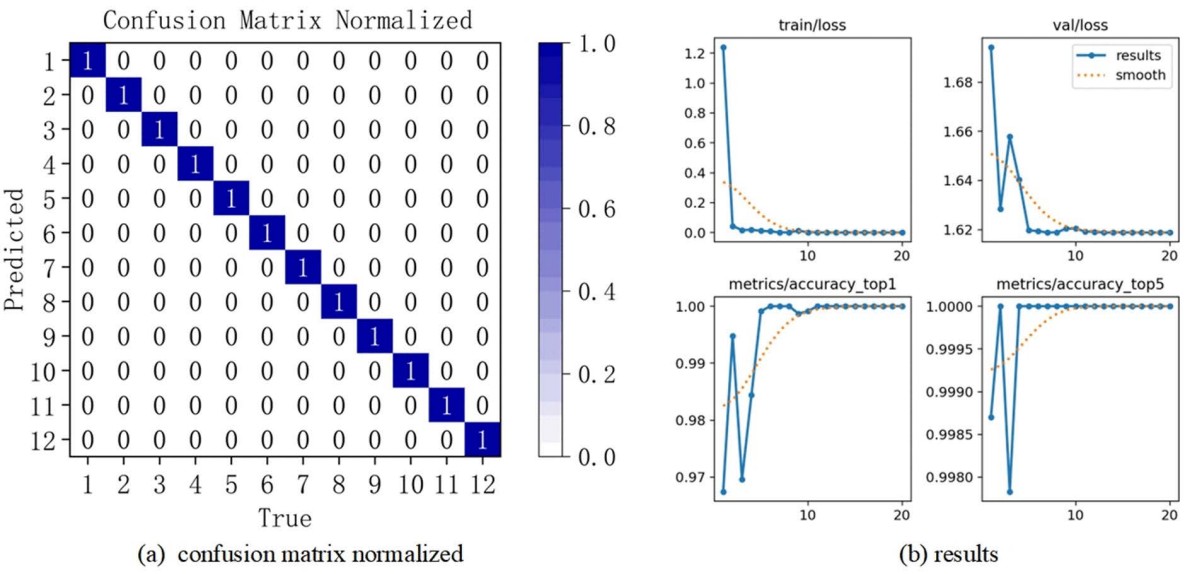

Fig 12. Bearing fault diagnosis confusion matrix.

Table 2. Diagnostic accuracy of the suggested method and other methods.

| Models | Sample Type | Average accuracy (%) |
|---|---|---|
| CNN-HMM [34] | Raw Data | 98.125 |
| TTLN [35] | Raw Data | 98.9 |
| MTSTLF [36] | Raw Data | 95.63 |
| ResNet-SVM [37] | Time-frequency image | 98.75 |
| DGNN [38] | Frequency domain signal | 97.81 |
| DenseNet-SVM [22] | Time-frequency image | 99.06 |
| YOLO v8 | Time-frequency image | 99.80 |
| Proposed model | Time-frequency image | 100 |

**4.1.4. Paderborn university bearing data set.** The Paderborn University bearing dataset is also chosen in this research to further confirm the effectiveness of the suggested approach [39]. The dataset functions under the following circumstances and has a sampling frequency of 64 kHz: 1500 revolutions per minute of rotation, a radial push of 400 Newtons, and a load moment of 0.7 Newton meters. The 120,000 data points that make up the original vibration signals are processed using a sample length of 1000 data points and a 500 data point overlap between neighboring sections. The health state data of each category is divided according to different damage modes, resulting in 239 groups for each category of damage and fault data. Following that, each image category is divided into 8:2 training and test samples. Lastly, as indicated in Table 3, the data files from every category are combined to create a sample set of 2151 time-frequency pictures.

Using the YOLO v8-C-OD fault diagnosis model constructed in this paper, the training samples obtained from each of the five types of health conditions constructed above are trained, and then the best model derived from the training is utilized for the test samples to obtain the final fault diagnosis accuracy, and the resulting confusion matrix and results are presented in Fig 13 (a)(b).

**Table 3. Description of the Paderborn University dataset sample types.**

| Fault types | Fault location | Number of training/test samples | Number of sample points | Lable |
|---|---|---|---|---|
| Normal (Nor) | Normal | 192/47 | 1000 | Class0 |
| Drilling (OD) | Bearing outer ring | 192/47 | 1000 | Class1 |
| electric engraver(E) | Bearing inner ring | 192/47 | 1000 | Class2 |
| | Bearing outer ring | 192/47 | 1000 | |
| Sharp trench by ED(EDM) | Bearing inner ring | 192/47 | 1000 | Class3 |
| | Bearing outer ring | 192/47 | 1000 | |
| Plastic deform (OP) | Bearing outer ring | 192/47 | 1000 | Class4 |
| Fatigue: pitting (FP) | Bearing inner ring | 192/47 | 1000 | Class5 |
| | Bearing outer ring | 192/47 | 1000 | |

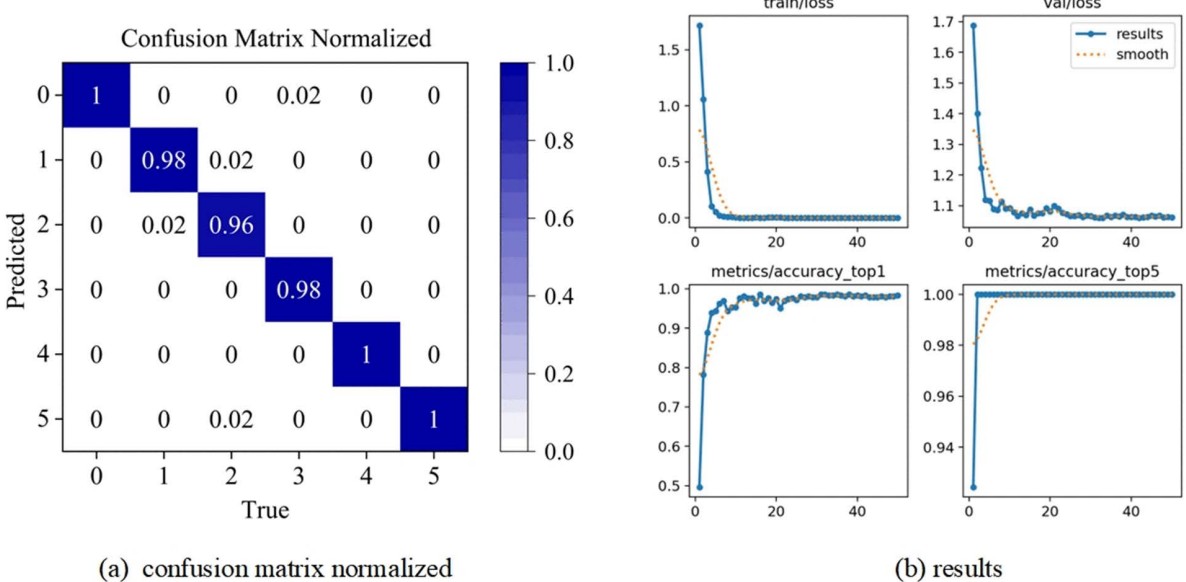

(a) confusion matrix normalized (b) results

**Fig 13. Paderborn University Bearing Fault Diagnosis Confusion Matrix.**

Comparison experiments are conducted with the BN, CNN, LSTM, V-BN, and K2-BN methods in the literature [40], which also use the Paderborn University bearing dataset under the same conditions, the findings are illustrated in Fig 14. The effectiveness of the suggested approach is confirmed by the experimental results, which clearly show that it performs better than the other four methods. This suggests that the proposed approach could be used effectively to diagnose rolling bearing faults in rotating machinery components.

### 4.2. Gear fault diagnosis

**4.2.1. Experimental data set.** The QPZZ-II rotating equipment vibration analysis and fault diagnosis testing platform system of Qianpeng Company [41] provided the data used in this investigation. The test platform is presented in Fig 15, which is mainly constituted of a variable-speed drive motor, bearings, a gearbox, two bias turntables, a governor, and other components. Both large and tiny gears are housed in the gearbox; the large gear's module has two teeth, 75 total, and is made of S45C. Likewise, the pinion gear's module has two teeth, 55 total, and is made of S45C. 5.12 kHz is the

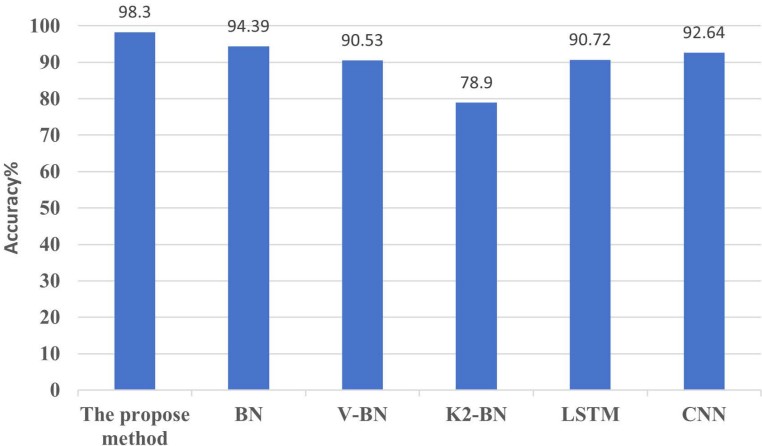

**Fig 14. Diagnostic accuracy of different models under the Paderborn University dataset.**

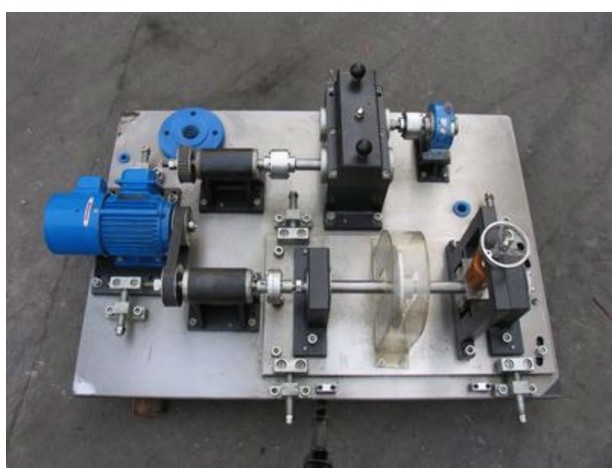

**Fig 15. Gearbox test platform.**

vibration signals' sampling rate. Both the input and output shafts' bearing blocks have acceleration sensors installed on them. The test simulates five fault states of the gearbox, including large gear pitting, large gear breaking, pinion gear wear, a combined fault of large gear breaking and pinion gear wear, and a combined fault of large gear pitting and pinion gear wear.

In this thesis, the experimental data are selected from the gearbox vibration signal data under the no-load condition of input speed $n = 880$ r/min, the normal state, and three types of single fault states. Since the acceleration monitoring data of this test platform system in the Y direction is more obvious, the vibration signal in the Y direction is selected for experimental testing in this paper.

**4.2.2. Sample description.** In this experiment, the data from the selected four health conditions, including the no-fault state, are used as the source of the experimental data, and 51,200 data points are extracted from the data in each health condition, which are used to make the samples. When the sample length of a single sample is too short, it will not contain enough fault information, resulting in one-sided fault diagnosis results that are not representative, and if the

sample duration is overly long, it will result in feature repetition, and the subsequent time-frequency conversion will result in a smaller target of fault feature information, which makes it easy to lose the fault information when diagnosing and recognizing and make the final fault diagnosis results inaccurate, so we choose 2048 data points as the sample length. The overlap of 1024 data points between neighboring parts is divided, and the data under each type of health condition is divided into 49 groups. After continuous wavelet transform (CWT), 49 time-frequency images of vibration signals under each health state are obtained. Then, in an 8:2 ratio, each kind of image is separated into two sections: training samples and test samples. As indicated in Table 4, the data files of all kinds are eventually combined to create a sample set with 196 time-frequency images. The time-frequency pictures of the four health statuses listed in Table 4 are shown sequentially in Fig 16(a)-(d).

**4.2.3. Comparative analysis of diagnostic results.** First, we train the training samples from each of the four categories of health issues listed above independently using the YOLO v8-C-OD fault diagnostic model built into this research. To attain the highest fault diagnosis accuracy, the performance of the best fault diagnostic model developed is then evaluated using test samples. The resulting confusion matrix and results are presented in Fig 17 (a)(b).

Again, the experiment was repeated 10 times, and the classification accuracy of these 10 experiments is displayed in Fig 18. The average diagnostic accuracy was 99.75%, with a maximum diagnostic accuracy reaching 100%. This indicates the robust feature learning and fault diagnosis capabilities of the deep model proposed in this paper.

We contrast the suggested approach with the conventional YOLO v8 fault diagnosis classification model to confirm its superiority for gearbox fault diagnosis. Table 5 shows the combined fault diagnostic findings for both models.

It is evident that the suggested approaches achieve convergence more quickly and have cumulative fault diagnostic accuracies that are all greater than the typical YOLO v8 model. The number of training iteration steps is lower when compared to the normal YOLO v8 model, suggesting that the YOLO v8-C-OD model developed in this article has a much better fault diagnostic impact. This further demonstrates that the suggested approach may partially solve the issues of information loss and the challenge of identifying tiny targets during the feature extraction process of fault signals.

To more forcefully illustrate the superiority of the proposed model in the domain of gearbox fault diagnosis, we also made a comparison between it and the existing methods such as BPNN, PNN [42], GWO-KELM [43], SDAE [44], and

**Table 4. Gearbox sample categories description.**

| Fault types | Measured speed/(r·min⁻¹) | Number of training/test samples | Sample points | Lable |
|---|---|---|---|---|
| Normal | 880 | 39/10 | 2048 | Class1 |
| Large Gear Pitting | 880 | 39/10 | 2048 | Class2 |
| Broken teeth on large gear | 878 | 39/10 | 2048 | Class3 |
| Pinion wear | 881 | 39/10 | 2048 | Class4 |

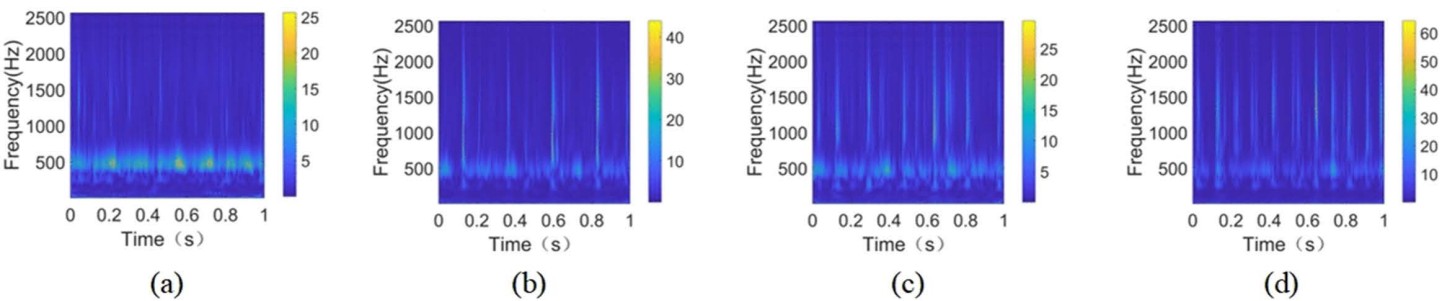

**Fig 16. Time-frequency images of four healthy states on Qianpeng dataset.** (a) Class 1; (b) **Class 2;** (c) Class 3; (d) **Class 4.**

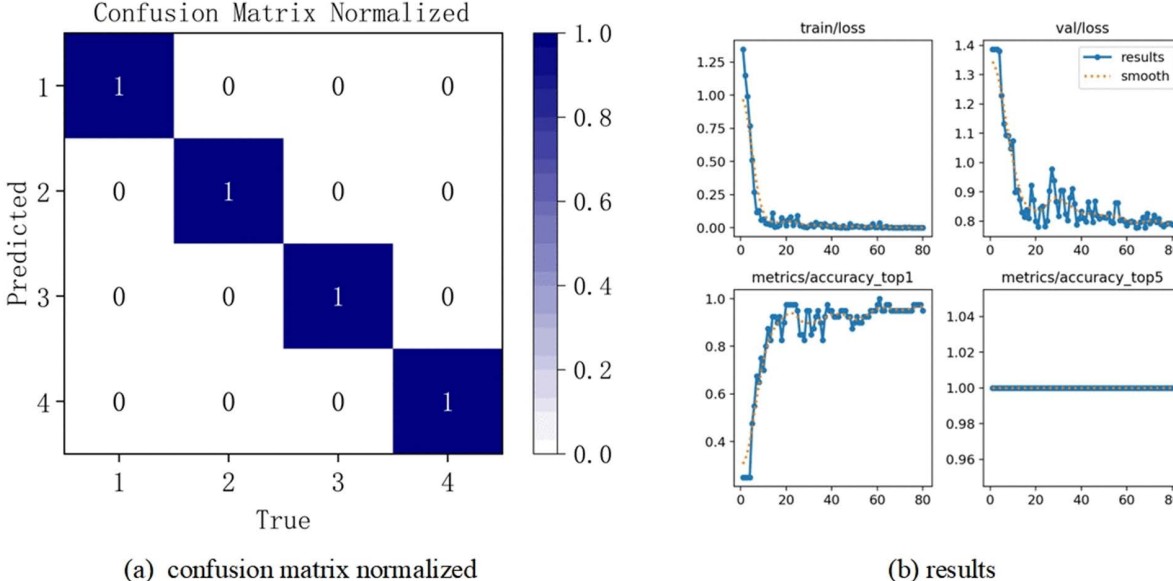

(a) confusion matrix normalized

(b) results

**Fig 17. Confusion matrix for gear troubleshooting.**

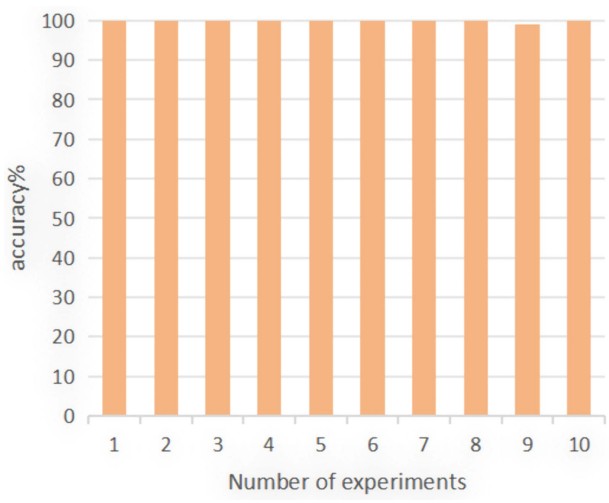

**Fig 18. Fault diagnosis result.**

DenseNet-SVM [22]. As in this experiment, each method is validated using the gearbox fault dataset of Jiangsu Qianpeng Diagnostic Engineering Co. In the literature [42], for both BP neural network (BPNN) and probabilistic neural network (PNN) methods, the normal state, tooth root crack, and broken tooth state of the gearbox are classified by inputting the probabilistic neural network model after data have processed the vibration signals to obtain sample data. In literature [43], a fault classification identification model based on the Gray Wolf Optimization Kernel Extreme Learning Machine (GWO-KELM) was established through the calculation of the time-domain feature parameters of the original vibration signals and performing information fusion. In literature [44], a gear condition monitoring model was established by extracting multi-sensor vibration time domain signals for data-level fusion and utilizing information fusion and stack noise reduction

**Table 5. The diagnostic accuracy and training duration of each model.**

| Model | Average accuracy (%) | Average number of training iteration steps/times |
|---|---|---|
| YOLO v8-C-OD model | 99.75 | 100 |
| Standard YOLO v8 model | 97.50 | 120 |

self-coding (SDAE) for layer-by-layer feature extraction. In the literature [22], the goal of fault diagnosis is achieved by first extracting the fault features using a continuous wavelet transform, then converting the original signal to the time-frequency domain, and thoroughly extracting the fault features using the DenseNet-SVM fault diagnostic model. The diagnostic accuracies of the five methods are listed in Table 6, which are 89.06%, 96.88%, 95.58%, 95.17%, and 99.68% respectively.

Table 6 shows that compared to other techniques mentioned in the sources, the YOLO v8-C-OD fault detection approach suggested in this study achieves greater diagnostic accuracy. This offers more proof that the suggested approach is more accurate in identifying gearbox problems.

## 4.3. Ablation experiment

Based on the original YOLO v8 model, this section evaluates the specific impact of the CBAM and ODconv components on model classification accuracy through ablation experiments. Consistent with previous experimental settings, each set of experiments was repeated 10 times to ensure accuracy. The mean, standard deviation, and relevant experimental data for each set of experiments conducted on different datasets using the proposed model are listed in Table 7. The fluctuation range of the model's fault diagnosis accuracy was controlled within ±0.75, indicating the model's stability across different datasets. The ablation experiment results are shown in Table 8, conducted under three dataset conditions. Table 8 shows that CBAM improved classification accuracy by 0.11%, 0.7%, and 1.25% across different datasets. Meanwhile, the parameter GFLOPs (gigaflops per second) indicates that ODconv not only enhanced model performance (increases of 0.1%, 1.1%, and 1%) but also significantly reduced computational complexity. This demonstrates that both CBAM and ODconv effectively enhance model classification performance, with ODconv additionally offering the advantage of reduced computational complexity. When synergistically applied, the model achieves optimal classification accuracy, fully validating their collaborative effect.

To further validate the model's accuracy and stability, we designed a cross-validation testing scheme. During implementation, the following operations were performed on different datasets: a number of time-frequency image samples equal to the test set were randomly sampled from each training set, and these samples were swapped with corresponding samples in the test set. After data reorganization, the model training and validation process was re-executed. The final cross-validation results are detailed in Table 9.

**Table 6. Diagnostic result of each model.**

| Categories | Methods | Average accuracy (%) |
|---|---|---|
| Shallow learning | BPNN [42] | 89.06 |
| | PNN [42] | 96.88 |
| | GWO-KELM [43] | 95.58 |
| Deep learning | SDAE [44] | 95.17 |
| | DenseNet-SVM [22] | 99.68 |
| | YOLO v8 | 97.50 |
| | The proposed method | 99.75 |

**Table 7. Diagnostic accuracy, mean, and standard deviation of the YOLO v8-C-OD model across 10 experiments.**

| Method | Dataset | Diagnostic Accuracy Rate per Experiment(%) | | | | | | | | | | Mean(%) | Standard deviation |
|---|---|---|---|---|---|---|---|---|---|---|---|---|---|
| | | 1 | 2 | 3 | 4 | 5 | 6 | 7 | 8 | 9 | 10 | | |
| YOLO v8-C-OD | CWRU Dataset | 100 | 100 | 100 | 100 | 100 | 100 | 100 | 100 | 100 | 100 | 100 | 0 |
| | Paderborn University Dataset | 98.3 | 98.3 | 98.3 | 98.3 | 97.5 | 98.3 | 99.1 | 98.3 | 98.3 | 98.3 | 98.3 | 0.36 |
| | Gearbox Dataset | 100 | 100 | 100 | 100 | 100 | 100 | 100 | 100 | 97.5 | 100 | 99.75 | 0.75 |

**Table 8. Ablation experiment results for model classification accuracy across different datasets.**

| Dataset | CBAM | ODconv | Diagnostic average accuracy(%) | GFLOPs | epoch |
|---|---|---|---|---|---|
| CWRU Dataset | × | × | 99.80 | 3.3 | 50 |
| | √ | × | 99.91 | 3.3 | 50 |
| | × | √ | 99.90 | 2.6 | 50 |
| | √ | √ | **100** | **2.6** | **50** |
| Paderborn University Dataset | × | × | 96.50 | 3.3 | 80 |
| | √ | × | 97.20 | 3.3 | 80 |
| | × | √ | 97.60 | 2.6 | 80 |
| | √ | √ | **98.30** | **2.6** | **80** |
| Gearbox Dataset | × | × | 97.50 | 3.3 | 100 |
| | √ | × | 98.75 | 3.3 | 100 |
| | × | √ | 98.50 | 2.6 | 100 |
| | √ | √ | **99.75** | **2.6** | **100** |

**Table 9. Cross-validation test results of the YOLO v8-C-OD model across different datasets.**

| Dataset | Average accuracy (%) | Average number of train-ing iteration steps/times |
|---|---|---|
| CWRU Dataset | 99.93 | 50 |
| Paderborn University Dataset | 98.60 | 80 |
| Gearbox Dataset | 99.50 | 100 |

Analysis of the experimental data reveals that the cross-validation results exhibit high consistency with the previous experimental outcomes. Specifically, across multiple datasets, the fluctuation range of fault diagnosis accuracy is consistently controlled within ±0.3%. This result indirectly confirms the robustness of the model architecture while validating the reproducibility of the experimental workflow and the reliability of the outcomes. Through systematic data permutation verification, we have both eliminated the random errors inherent in single-run experiments and ensured the model's generalization capability across different data distributions, providing a reliable theoretical foundation for subsequent practical applications.

## 5. Conclusion

This paper examines the existing research on fault diagnosis techniques for rotating mechanical component systems and discusses the necessity of fault diagnosis for rotating mechanical components. Aiming at the problems of easy loss of information and insufficient processing of the existing depth feature extraction methods, a fault diagnosis method for rotating mechanical components based on improving YOLO v8 and fusing CBAM is suggested. According to experiments, the method in this research can achieve 100% diagnostic precision. for the 12 health conditions selected in the CWRU bearing fault dataset and 99.75% for the 4 health conditions selected in the gearbox fault dataset of Qianpeng Company.

which also has a big advantage over other bearing failure diagnostic models using the Paderborn University dataset. The method is validated for its effectiveness, stability, and superiority.

## Author contributions

**Conceptualization:** xiaoguang wang.

**Methodology:** xiaoguang wang, laohu yuan, le ma, jiafu liu.

**Project administration:** laohu yuan.

**Software:** xiaoguang wang, le ma.

**Supervision:** laohu yuan, jiafu liu.

**Writing – original draft:** xiaoguang wang.

**Writing – review & editing:** laohu yuan, jiafu liu.

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
