## [Decision Letter · Decision Letter 0]

9 Sep 2025

PONE-D-25-42680A fault diagnosis method for  rotating machinery components based on enhanced YOLO v8 and integrated attention mechanismPLOS ONE

Dear Dr. yuan,

Thank you for submitting your manuscript to PLOS ONE. After careful consideration, we feel that it has merit but does not fully meet PLOS ONE’s publication criteria as it currently stands. Therefore, we invite you to submit a revised version of the manuscript that addresses the points raised during the review process.

We look forward to receiving your revised manuscript.

Kind regards,

Dandan Peng

Academic Editor

PLOS ONE

Journal Requirements:

Reviewers' comments:

Reviewer's Responses to Questions

**Comments to the Author**

1. Is the manuscript technically sound, and do the data support the conclusions?

Reviewer #1: Partly

Reviewer #2: Yes

2. Has the statistical analysis been performed appropriately and rigorously? 

Reviewer #1: Yes

Reviewer #2: Yes

3. Have the authors made all data underlying the findings in their manuscript fully available?

Reviewer #1: Yes

Reviewer #2: Yes

4. Is the manuscript presented in an intelligible fashion and written in standard English?

Reviewer #1: Yes

Reviewer #2: Yes

5. Review Comments to the Author

Reviewer #1: The manuscript requires further validation through statistical analysis, ablation studies, model complexity evaluation, and broader generalization experiments before it can be considered for publication.

Specific revision suggestions are detailed in the attachment.

Reviewer #2: Dear Authors

I have carefully reviewed the manuscript entitled “A fault diagnosis method for rotating machinery components based on enhanced YOLO v8 and integrated attention mechanism”, submitted for consideration inPLOS ONE. The article may be of interest to the solid mechanics community. The fault diagnosis model for rotating mechanical components proposed in the manuscript is of great significance to fields such as industrial equipment maintenance and safe production. However, there are several points that require improvement or clarification before the manuscript can meet the standards expected for publication in the journal.

Please find my detailed comments below:

1. Please modify the format of the paper according to the submission requirements of PLOS ONE, such as aligning the paragraphs at both ends.

2. Could the author explain why the YOLO v8 model has already achieved high prediction accuracy in Tables 2, 5, and 6 for equipment fault detection, but the YOLO v8-C-OD fault diagnosis model is still proposed? What advantages does the proposed model have?

3. Is the sample set in the YOLO v8-C-OD fault diagnosis model sufficient, and is the data collection method scientific?

4. Has the proposed YOLO v8-C-OD fault diagnosis model been used with self-collected data to predict faults in rotating mechanical components, to verify the accuracy and reliability of the proposed model? It is suggested to add actual cases or field test data.

6. PLOS authors have the option to publish the peer review history of their article (what does this mean? ). If published, this will include your full peer review and any attached files.

**Do you want your identity to be public for this peer review?** For information about this choice, including consent withdrawal, please see our Privacy Policy .

Reviewer #1: No

Reviewer #2: No

---

## [Author Response · Author response to Decision Letter 1]

7 Nov 2025

Reviewer #1:

We thank reviewer for all these valuable suggestions. We have corrected and improved the whole manuscript according to these suggestions point by point.

1.I don't understand why the article requires images and manuscripts to be uploaded separately. Why don't authors insert images into the manuscript to make it easier for reviewers and readers to read and study?

Answer: We thank reviewer for this comment, and sincerely thank you for your meticulous attention and valuable suggestions regarding the figure formatting in the manuscript. Due to the journal's suggestion that submitted manuscript files should not exceed 20MB, and the requirement that figures be uploaded as separate files, we uploaded the figures and text separately. We fully understand the inconvenience this may have caused you during the review process and sincerely apologize for this. We have incorporated all relevant figures into the corresponding locations within the main text as suggested, enabling you to intuitively reference the chart content while reviewing the textual logic.

2.The migration of the YOLO series object detection architecture to mechanical fault diagnosis is not without precedent. The authors need to define their contributions more precisely: is replacing the convolutional operators and attention modules alone sufficient to support the claim of a ‘novel fault diagnosis method’? To highlight the innovativeness of their paper, the authors need to provide further details.

Answer: We thank reviewer for this comment. This has prompted us to recognize more clearly the shortcomings in defining the paper's innovation.

(1) The point you raised—that “Transfer YOLO is not original”—is indeed the starting point we must confront. We do not pursue the ‘originality’ label, but rather aim to solidify the academic claim of a “novel fault diagnosis method” through rigorous details that make it both defensible and comprehensible. We can address this from three specific angles:

Regarding theoretical adaptability, Section 2.1 analyzed that while YOLO v8 enhances learning capacity by replacing C3 with C2f, subtle vibration fault features remain highly elusive and prone to omission during feature extraction—particularly evident in small object detection. Furthermore, Section 2.2 introduced the CBAM attention mechanism, which integrates channel and spatial attention to focus on critical information, reduce emphasis on secondary features, and improve fault feature extraction rates. Section 2.3 introduced ODConv, which dynamically adjusts convolution kernel weights. While maintaining low computational overhead and parameter count, it significantly enhances fault feature extraction capability, demonstrating outstanding performance in small object detection and fine-grained classification tasks. Theoretical derivation indicates that integrating CBAM and ODConv effectively improves YOLO v8's fault detection performance.

Experimental validation involves comparisons with classical methods (e.g., MTSTLF, DGNN) and similar transfer methods (e.g., CNN-HMM) on the same dataset. Comparative results demonstrate the superiority of the proposed method.

In terms of contribution quantification, we supplement Section 4.3 with ablation experiments. Using the controlled variable method, we validate that: introducing CBAM alone enhances feature focusing capability and improves model diagnostic accuracy; introducing ODConv alone optimizes weight allocation efficiency, boosts model diagnostic accuracy, and reduces computational complexity; while the synergistic effect of both achieves a performance leap.

(2) We further validate the contributions of CBAM and ODConv modules to the model, as well as their synergistic effect in achieving a performance leap for the YOLO v8 model, through supplementary ablation experiments in Section 4.3. The specific supplementary experimental content is as follows:

The ablation experiment results are shown in Table 8, conducted under three dataset conditions. Table 8 shows that CBAM improved classification accuracy by 0.11%, 0.7%, and 1.25% across different datasets. Meanwhile, the parameter GFLOPs (gigaflops per second) indicates that ODconv not only enhanced model performance (increases of 0.1%, 1.1%, and 1%) but also significantly reduced computational complexity. This demonstrates that both CBAM and ODconv effectively enhance model classification performance, with ODconv additionally offering the advantage of reduced computational complexity. When synergistically applied, the model achieves optimal classification accuracy, fully validating their collaborative effect.

Table 8. Ablation experiment results for model classification accuracy across different datasets.�The specific data are listed in the Response to Reviewers document and in the tables within the manuscript

3.The results are too perfect and lack statistical validation. Some of the results show a diagnostic accuracy of 100%, which is difficult to achieve in complex working conditions. Rather than simply providing the average accuracy rate, it is recommended that the authors supplement the mean and standard deviation of multiple experiments. Cross-validation or confidence intervals could also be added to improve the credibility of the results.

Answer: We thank reviewer for this comment. The challenge you highlighted—that “100% accuracy is difficult to achieve under complex operating conditions”—is precisely the reality we must confront. We fully acknowledge this suggestion and explicitly state that the 100% fault diagnosis rate reported for the Western Reserve University dataset is based solely on the current experimental test set results, which inherently lack validation for operational condition generalization. Accordingly, following your recommendation, we have supplemented Section 4.3 with statistical data from multiple repeated experiments conducted after the ablation tests. Additionally, we have introduced a cross-validation mechanism to redistribute the dataset for training and testing, thereby validating the model's performance from both statistical stability and generalization capability perspectives. The specific modifications are as follows:

Based on the original YOLO v8 model, this section evaluates the specific impact of the CBAM and ODconv components on model classification accuracy through ablation experiments. Consistent with previous experimental settings, each set of experiments was repeated 10 times to ensure accuracy. The mean, standard deviation, and relevant experimental data for each set of experiments conducted on different datasets using the proposed model are listed in Table 7. The fluctuation range of the model's fault diagnosis accuracy was controlled within ±0.75, indicating the model's stability across different datasets.

Table 7. Diagnostic accuracy, mean, and standard deviation of the YOLO v8-C-OD model across 10 experiments.�The specific data are listed in the Response to Reviewers document and in the tables within the manuscript

To further validate the model's accuracy and stability, we designed a cross-validation testing scheme. During implementation, the following operations were performed on different datasets: a number of time-frequency image samples equal to the test set were randomly sampled from each training set, and these samples were swapped with corresponding samples in the test set. After data reorganization, the model training and validation process was re-executed. The final cross-validation results are detailed in Table 9. Cross-validation test results of the YOLO v8-C-OD model across different datasets.�The specific data are listed in the Response to Reviewers document and in the tables within the manuscript

Analysis of the experimental data reveals that the cross-validation results exhibit high consistency with the previous experimental outcomes. Specifically, across multiple datasets, the fluctuation range of fault diagnosis accuracy is consistently controlled within ±0.3%. This result indirectly confirms the robustness of the model architecture while validating the reproducibility of the experimental workflow and the reliability of the outcomes. Through systematic data permutation verification, we have both eliminated the random errors inherent in single-run experiments and ensured the model's generalization capability across different data distributions, providing a reliable theoretical foundation for subsequent practical applications.

4.The ablation experiments in this paper are inadequate. Although there are some comparative experiments, the benefits of ODConv and CBAM have not been clearly distinguished. It is therefore recommended that the authors supplement the paper with explanations of each module's contribution.

Answer: We thank reviewer for this comment. This has deepened our understanding of the critical importance of distinguishing module contributions in ablation experiments. The point you raised—that “the advantages of ODConv and CBAM were not clearly differentiated”—indeed represents a key shortcoming in the original experimental design. While our focus was on validating the effectiveness of module combinations, we failed to sufficiently deconstruct the independent contribution mechanisms of individual modules.

To address this, we have conducted a dedicated supplementary ablation study in Section 4.3 to observe specific changes in model performance when individual modules are removed. Quantitative comparisons confirm that both CBAM and ODConv enhance the fault detection accuracy of the YOLO v8 model, while ODConv effectively reduces computational complexity. Their synergistic interaction achieves a significant performance leap. The specific modifications are as follows:

The ablation experiment results are shown in Table 8, conducted under three dataset conditions. Table 8 shows that CBAM improved classification accuracy by 0.11%, 0.7%, and 1.25% across different datasets. Meanwhile, the parameter GFLOPs (gigaflops per second) indicates that ODconv not only enhanced model performance (increases of 0.1%, 1.1%, and 1%) but also significantly reduced computational complexity. This demonstrates that both CBAM and ODconv effectively enhance model classification performance, with ODconv additionally offering the advantage of reduced computational complexity. When synergistically applied, the model achieves optimal classification accuracy, fully validating their collaborative effect.

Table 8. Ablation experiment results for model classification accuracy across different datasets.�The specific data are listed in the Response to Reviewers document and in the tables within the manuscript.

These supplementary experiments not only address your concerns regarding the “contribution mechanism” but also provide us with a clearer understanding of the interaction principles between modules. We fully recognize that the rigor of scientific research lies in the pursuit and verification of details. Therefore, we earnestly request reviewers to continue pointing out our shortcomings and assist us in further refining this study.

5.Additionally, there are significant issues with the paper's formatting, citations and image resolution. For instance, the explanations of the symbols in equations 3, 4 and 6 are incomplete. It is recommended that the physical meanings of the variables are supplemented. The citation format is inconsistent. The resolution of Figures 8, 11 and 16 is slightly low. The authors are recommended to systematically revise these issues.

Answer: We thank reviewer for this comment. Your keen observation regarding the missing symbol explanations in Equations 3, 4, and 6, the inconsistencies in citation formatting you pointed out, and the issues with insufficient resolution in Figures 8, 11, and 16 are invaluable to us.

(1)Improvements to the Explanation of Equation Symbols: We have reviewed each variable symbol individually and added corresponding explanatory notes for each one. The specific modifications are as follows:�For specific modifications, please refer to the “Response to Reviewer Comments” and the manuscript.

(2)Regarding the standardization of citation formats: This was indeed due to our oversight. We meticulously reviewed each entry in the reference list, strictly reformatted them according to the journal's requirements, and corrected all inconsistencies in author name capitalization and journal abbreviations. We have made the following modifications:

For example

Reference [3]

[3] Zhao Y, Wang J, Li X, Peng G, Yang Z. Extended least squares support vector machine with applications to fault diagnosis of aircraft engine. Isa Transactions. 2020;97:189-201.

Reference [6]

[6] Chen P. Review of fault diagnosis methods for rolling bearings based on vibration signals. Bearings. 2022(06):1-6.

(3) Regarding the enhancement of image resolution: We have completely remastered these images and increased their resolution to over 600 dpi to ensure precise visibility in video footage. Additionally, the corresponding images have been embedded at their respective locations within the main text of the paper.

Reviewer #2:

We thank reviewer for all these valuable suggestions. We have corrected and improved the whole manuscript according to these suggestions point by point.

1.Please modify the format of the paper according to the submission requirements of PLOS ONE, such as aligning the paragraphs at both ends.

Answer: We thank reviewer for this comment. This provides important guidance for enhancing the academic rigor and readability of our paper. Regarding the formatting issues you pointed out, we have thoroughly reviewed and revised the manuscript in strict accordance with PLOS ONE's submission requirements, including aligning paragraphs to both margins and adjusting line spacing.

2.Could the author explain why the YOLO v8 model has already achieved high prediction accuracy in Tables 2, 5, and 6 for equipment fault detection, but the YOLO v8-C-OD fault diagnosis model is still proposed? What advantages does the proposed model have?

Answer: We thank reviewer for this comment. This prompts us to reflect more deeply on the necessity of model refinement.

(1)The observation that “YOLO v8 achieves high accuracy in Tables 2, 5, and 6” indeed served as the starting point for our work. However, these very datasets revealed the limitations of the original model—while demonstrating excellent overall performance, it still showed room for improvement in complex scenarios such as detecting minor vibration faults and extracting features from small objects.

Specifically, while the original YOLO v8 model achieves high-precision diagnostics under standard conditions, its C2f architecture risks “over-smoothing” during feature extraction when confronting the “feature concealment” challenge common in industrial settings—such as the faint vibration signals generated by early-stage equipment failures—thereby weakening critical fault signatures. This is precisely the core motivation behind our proposed YOLO v8-C-OD model: integrating the CBAM attention mechanism with the ODConv dynamic convolution module to achieve a significant leap in overall performance. We do not dismiss the value of the original model but aim to make the fault diagnosis model more aligned with industrial practical needs through this “precision enhancement.” We supplemented Section 4.3 with ablation experiments and quantitatively validated the independent contributions and synergistic effects of each module.

(2)The new model's advantages manifest across three dimensions: First, CBAM precisely localizes fault feature regions through dual channel-spatial attention, reducing focus on secondary features and enhancing fault feature extraction rates; Second, ODConv dynamically adjusts convolution kernel weights to significantly enhance small object detection capabilities while maintaining low computational overhead. Third, the synergistic interaction between these two components forms a positive feedback loop of “feature enhancement-weight adaptation,” substantially improving the model's fault detection performance across operating conditions and in small-object scenarios. We kindly request reviewers to continue focusing on refining these details to help us enhance the practical guidan

---

## [Decision Letter · Decision Letter 1]

24 Nov 2025

A fault diagnosis method for  rotating machinery components based on enhanced YOLO v8 and integrated attention mechanism

PONE-D-25-42680R1

Dear Dr. yuan,

We’re pleased to inform you that your manuscript has been judged scientifically suitable for publication and will be formally accepted for publication once it meets all outstanding technical requirements.

Kind regards,

Dandan Peng

Academic Editor

PLOS ONE

Additional Editor Comments (optional):

Reviewers' comments:

Reviewer's Responses to Questions

**Comments to the Author**

1. If the authors have adequately addressed your comments raised in a previous round of review and you feel that this manuscript is now acceptable for publication, you may indicate that here to bypass the “Comments to the Author” section, enter your conflict of interest statement in the “Confidential to Editor” section, and submit your "Accept" recommendation.

Reviewer #1: All comments have been addressed

Reviewer #2: All comments have been addressed

2. Is the manuscript technically sound, and do the data support the conclusions?

Reviewer #1: Yes

Reviewer #2: Yes

3. Has the statistical analysis been performed appropriately and rigorously? 

Reviewer #1: Yes

Reviewer #2: Yes

4. Have the authors made all data underlying the findings in their manuscript fully available?

Reviewer #1: Yes

Reviewer #2: Yes

5. Is the manuscript presented in an intelligible fashion and written in standard English?

Reviewer #1: Yes

Reviewer #2: Yes

6. Review Comments to the Author

Reviewer #1: (No Response)

Reviewer #2: The authors made extensive revisions to the paper. It would be great if the author could continue to patiently check the details such as English grammar, formulas and pictures before publishing the paper.

7. PLOS authors have the option to publish the peer review history of their article (what does this mean? ). If published, this will include your full peer review and any attached files.

**Do you want your identity to be public for this peer review?** For information about this choice, including consent withdrawal, please see our Privacy Policy .

Reviewer #1: No

Reviewer #2: No

---

## [Editor Report · Acceptance letter]

PONE-D-25-42680R1

PLOS ONE

Dear Dr. yuan,

I'm pleased to inform you that your manuscript has been deemed suitable for publication in PLOS ONE. Congratulations! Your manuscript is now being handed over to our production team.

Kind regards,

on behalf of

Dr. Dandan Peng

Academic Editor

PLOS ONE